# A Survey on the Knowledge, Attitudes, and Practices of Lebanese Physicians Regarding Air Pollution

**DOI:** 10.3390/ijerph19137907

**Published:** 2022-06-28

**Authors:** Hazem I. Assi, Paul Meouchy, Ahmad El Mahmoud, Angela Massouh, Maroun Bou Zerdan, Ibrahim Alameh, Nathalie Chamseddine, Houry Kazarian, Salah Zeineldine, Najat A. Saliba, Samar Noureddine

**Affiliations:** 1Department of Internal Medicine, Naef K. Basile Cancer Institute, American University of Beirut Medical Center, P.O. Box 11-0236, Beirut 1107, Lebanon; ha157@aub.edu.lb (H.I.A.); pe10@aub.edu.lb (P.M.); ae155@aub.edu.lb (A.E.M.); maroun.zerdan@gmail.com (M.B.Z.); ia84@aub.edu.lb (I.A.); nc39@aub.edu.lb (N.C.); 2Hariri School of Nursing, American University of Beirut, P.O. Box 11-0236, Beirut 1107, Lebanon; am50@aub.edu.lb (A.M.); hk87@aub.edu.lb (H.K.); 3Department of Internal Medicine, Pulmonary and Critical Care Division, American University of Beirut Medical Center, P.O. Box 11-0236, Beirut 1107, Lebanon; sz01@aub.edu.lb; 4Department of Chemistry, Faculty of Arts and Sciences, American University of Beirut, P.O. Box 11-0236, Beirut 1107, Lebanon; ns30@aub.edu.lb

**Keywords:** air pollution, physicians, Lebanon

## Abstract

Introduction: Air pollution imposes a significant burden on public health. It is emerging as a modifiable risk factor for cancer, diabetes, and respiratory and cardiovascular diseases. This study aims to assess the knowledge, attitudes, and practices of Lebanese physicians regarding air pollution. Methods: This observational study uses a descriptive cross-sectional correlational design. The data were collected using a self-administered online survey that was sent to 874 potential respondents who are members of the Lebanese Order of Physicians. Data analysis was done using descriptive statistics and a chi-square test. Results: The results show a deficiency in the knowledge of physicians regarding many sources of air pollution, including dust, the smell of perfume, candles, vacuum cleaners, air fresheners, electronic cigarettes, etc. The majority of physicians agree that air pollution increases the risk of several health problems. Only 38% of physicians routinely ask their patients about exposure to air pollution, and 75% of them believe that they have a role as physicians in reducing air pollution levels. Over half of the sample are confident in counseling their patients on sources of air pollution, and two thirds of them are in support of including assessment of air pollution exposure during regular medical visits. Conclusion: Air pollution levels are progressively increasing over time. Given the health impact of exposure to air pollution, healthcare professionals need to stay up to date on this topic. The results of this study suggest the need for continuing education about air pollution for physicians and developing guidelines for what exactly to ask patients in assessing their exposure.

## 1. Introduction

Since the beginning of the industrial revolution, the world has witnessed a significant rise in air pollution levels. The World Health Organization (WHO) declared that 91% of the world’s population lives in regions where exposure to air pollution exceeds their guideline limits [1]. This is problematic because air pollution is a risk factor for the development of non-communicable diseases, such as cancer, diabetes mellitus, chronic respiratory diseases, stroke, and neurological conditions [2]. A multicenter project analyzing the data of 22 European cohorts showed that long-term exposure to fine particulate air pollution was associated with natural-cause mortality, even when the concentrations were below the mean limit value [2]. According to the WHO, 4.2 million deaths every year occur as a result of exposure to ambient air pollution, which is on par with mortality due to household air pollution (3.8 million premature deaths each year due to exposure to smoke from cooking fires) [1]. In a recent position statement, the American Heart Association reviewed the evidence on the link between exposure to particulate matter and air pollution and heart disease, specifying risk levels and proposing protective actions [3]. The Global Burden of Disease document associated ambient air pollution with 2.1 million deaths attributable to cardiovascular disease (myocardial infarction, strokes, and heart failure) [4]. Unfortunately, many healthcare professionals are not well informed about the dangers of air pollution. Rotily et al. found that the majority of physicians did not recognize air pollution as a direct cause of increased mortality and cardiovascular events [5]. Given its recognized status as a risk factor for many diseases and its strong association with increased mortality, it is crucial not only that physicians ask about and counsel their patients on this matter but also that they take it into consideration while formulating diagnoses and treatment plans.

Air pollution is assessed by measuring the amount of particulate matter (PM) and gaseous components in the air. PMs are defined as the solid and liquid substances that are present in the atmosphere, while gaseous components include carbon monoxide, volatile organic compounds (VOCs), nitrogen oxides, and ozone [6]. All of those compounds contribute to ambient air pollution in the outside environment. However, indoor air pollution is also on the rise due to the effects of smoking, organic solvents, heating, cooking, and other common household consumer products [7].

In Lebanon, the sources of air pollution are numerous: cars, diesel generators, waste burning, wildfires, industrial air emissions, etc. In 2012, a group of investigators measured the levels of particulate matter PM_2.5_ and PM_10_ in Beirut, documenting levels that exceed the annual WHO average limits by 150% and 200%, respectively. During the same time period, data were collected on 11,567 patients admitted to the emergency departments in seven hospitals in Beirut. The study evaluated the association between daily concentrations of particulate matter and emergency hospital admission for respiratory and cardiovascular illnesses. After adjusting for confounders, total respiratory admissions were significantly associated with increased levels of PM_10_ (1.012 [95% CI 1.004–1.02]) and PM_2.5_ (1.016 [95% CI 1.000–1.032]) recorded on the same day of those admissions [8]. Therefore, assessing physicians’ knowledge of air pollution is instrumental for the implementation of training programs and counseling workshops on its detrimental effects. To our knowledge, there is no study in the literature that addresses this topic in Lebanon nor in the Middle East and North Africa (MENA) region. The aim of our study is to investigate the knowledge, attitudes, and practices of Lebanese physicians related to air pollution.

## 2. Materials and Methods

Design: This was an observational study that used a descriptive cross-sectional correlational design. The data were collected using a self-administered online survey.

Sample: The sample consists of 125 physicians recruited from a target population of 874 physicians who are members in the following medical societies of the Lebanese Order of Physicians: Lebanese Society of Cardiology, Lebanese Society of Pulmonology, Lebanese Society of Oncology, Lebanese Society of Infectious Diseases, Lebanese Society of Family Medicine, Lebanese Society of General Practitioners, and the Lebanese Society of Internal Medicine. Recruitment was done between October 2020 and January 2021.

Instrument: The questionnaire used for the study was developed by a research team that included experts in air pollution, medicine, and survey development (See Appendix A). The questionnaire addresses knowledge of pollutants, sources of air pollution, living and working conditions associated with air pollution (questions 1–7), practices related to the discussion of air pollution with patients (questions 8–9), and attitudes related to the inclusion of air pollution in practice and training (questions 10–14). Most of the items are rated on five-point Likert scales, with the first two questions requiring yes/no answers. In addition, there were six demographic questions that address age, gender, years of experience, medical specialty, non-medical education, and location of practice.

Procedure: A pilot test of the questionnaire was conducted with five physicians who were not included in the main study to seek their opinion about the clarity of the questions. The officers of the medical societies listed above were then contacted to request a list of the email addresses of the members of those societies. The questionnaire was prepared on LimeSurvey and included an invitation script, consent form, and the survey questions. The questionnaire was sent to the email addresses provided and three reminder emails were sent at 2-week intervals to improve the response rate.

Statistics: Results are reported using descriptive statistics with absolute counts and percentages. A chi-square test was used to establish associations between demographics of the sample and their knowledge about air pollution. A *p*-value < 0.05 was considered significant. Statistical analysis was performed using the SPSS v. 27.0 statistical package.

## 3. Results

The sample included 125 physicians, which is a 14% response rate. Table 1 shows the demographic characteristics of the sample. The sample was almost equally distributed by gender and relatively young (only 33% were 50 years of age and older), and the most frequently represented specialties were family medicine and general practice, in the capital, Beirut (56%).

### 3.1. Knowledge of Pollutants and Sources of Air Pollution

As seen in Figure 1, there was significant discordance in the answers of doctors regarding humidity and dust storms. Approximately 20% believed humidity to be a source of pollution while almost 30% said they did not know whether it is or it is not. For urban gardens, 10.34% thought that they were sources of pollution while 11.21% did not know. The most noticeable discordance is in dust storms, where 27.50% said they were not sources of pollution and 20% did not know. For construction work, cars, incinerators, and diesel generators, most participants answered correctly, with at least 87% agreeing that they are sources of air pollution. Likewise, the majority of the participants believed that trees and urban gardens are not sources of air pollution.

In Figure 2, we can see that there was much conflict in the answers of participants. The only substance participants agreed upon almost unanimously was smoke, with 91.87% agreeing that it is indeed a source of pollution. Almost 50% believed dust to be a pollutant, while 39% said it was not, and 10% did not know. The smell of perfume and the smell of sewage were similarly discordant with 59.66% saying perfume was not a pollutant and 58.54% saying sewage was a pollutant, while for both approximately 20% did not know. For water vapor, about 72% believed it was not a pollutant, and 14.53% believed it was, while 13.68% were unsure about it.

Table 2 shows the results for which living conditions are believed to be associated with exposure to air pollution. There was an overwhelming agreement that living near a diesel generator, car or bus parking, or a garbage dump and smoking tobacco (arguileh and cigarettes) were all associated with exposure to air pollutants. There was some disagreement regarding exposure related to other living conditions such as living near an open well ventilated road intersection (60% agree, 20% disagree), living near a gas station (79% agree, 5% disagree), living near a construction site (80% agree, 8% disagree), living in a house with a garden (21% agree, 74% disagree), living near a garbage incinerator (82% agree, 9% disagree), staying indoors most of the time (24% agree, 31% disagree), living near a busy traffic intersection (84% agree, 9% disagree), smoking e-cigarettes (68% agree, 11% disagree), living on lower-level floors (45% agree, 9% disagree), living near the airport (72% agree, 10% disagree), living near an agricultural field (20% agree, 48% disagree), living with a smoker (84% agree, 9% disagree). It is worth noting the significant percent of participants who answered neutral to the following items: staying indoors most of the time (45.5%), living on lower-level floors (46.3%) and living near an agricultural field (32.2%).

According to Table 3, most doctors surveyed agreed that working at a gas station (86%), working with paints (85%), and working in a place where smoking is allowed (85%) are associated with exposure to air pollution. There was much contradiction regarding the other working conditions: working at a hospital (23% agree, 47% disagree), in an automobile repair shop (80% agree, 7% disagree), at a construction site (82% agree, 6% disagree), in a chemical lab (75% agree, 10% disagree), as a teacher (22% agree, 47% disagree), in a place where only vaping is allowed (61% agree, 9% disagree), as a farmer (31% agree, 43% disagree), in a place where smoking is not allowed (17% agree, 69% disagree), as a tour guide (25% agree, 34% disagree), at a clinic (19% agree, 50% disagree), and at a gardening shop (18% agree, 57% disagree). Moreover, at least one third of the sample were neutral on whether working as tour guide, as a teacher, at a clinic, at a hospital or at a place where only vaping is allowed are associated with exposure to air pollution.

When surveyed about household items that could be a source of pollution, opinions contrasted greatly about almost every item, as is evidenced in Table 4. The only item there was strong agreement on was indoor smoking, with almost 92% agreeing and 4% disagreeing that it is associated with air pollution. There was an 81%, 73%, 80%, 83%, 73%, 84%, and 82% consensus respectively about chimneys, wood heaters, detergents, pesticides and insecticides, mold and fungi on walls, lighting charcoal for arguileh at home, and diesel heaters indoors as potential correlates of air pollution at home. Answers about candles (47% agree, 17% disagree), vacuum cleaners (33% agree, 28% disagree), air fresheners (44% agree, 18% disagree), glues and adhesives (57% agree, 15% disagree), watching television (18% agree, 57% disagree), air condition unit (33% agree, 29% disagree), pets (31% agree, 39% disagree), indoor cooking (29% agree, 37% disagree), taking a shower (14% agree, 69% disagree), and open windows at home (20% agree, 52% disagree) were even more discordant. Moreover, one third of the sample was neutral on whether the following household items are linked to exposure to air pollution: candles, vacuum cleaners, pets, indoor cooking, air fresheners, glues/adhesives, and open windows at home.

Figure 3 illustrates that most doctors surveyed believed that air pollution substantially increases the risk of cancer (91%), COPD (89%), asthma (89%), and allergies (84%). There was also agreement, though to a lesser extent, of 72%, and 71% about increasing the risk of inability to breathe through the nose and prematurity and perinatal complications, respectively. There was less agreement, however, about CAD (60% increases, 13.11% does not increase), obesity (32% increases, 33% does not increase), and insomnia (48% increases, 17.8% does not increase). It is worth noting that one third of participants were not sure about the air pollution risk for coronary artery disease, insomnia, and obesity.

Finally, when asked whether increased air pollution levels are linked to the severity of an illness or disease, 76% of the physicians thought that increased air pollution is linked a lot or extremely to severity of illness.

### 3.2. Practices Related to the Discussion of Air Pollution with Patients

When asked how often they ask their patients about their exposure to air pollution, 6% answered always, 31.6% most of the time, 41% sometimes, and 21.4% rarely or not at all. The most frequently reported trigger for asking about exposure to air pollution was if the patient reported symptoms (31%), followed by having a case that is difficult to diagnose (12%), whereas 24.1% stated that they always ask about exposure to air pollution as part of their routine history taking.

### 3.3. Attitudes Related to Inclusion of Air Pollution in Practice and Training

The participants were asked how important it is for them to ask their patients about their exposure to air pollutants. The vast majority (83.3%) considered asking to be important (35.1%), very important (34.2%), or extremely important (14%).

In answer to the question “Do you think that you have a role in reducing air pollution?”, 75.2% answered that they have a role as physicians and 85% that they have a role as citizens. In terms of confidence in their ability to counsel their patients to reduce their exposure to air pollution, over half the sample reported being confident (31.3%), very confident (13.9%), or extremely confident (10.4%). Moreover, 72% expressed interest in attending continuing education about air pollution (36% interested, 26.3% very interested, and 9.6% extremely interested).

The last question addressed including an assessment of air pollution exposure during regular medical visits; two thirds of the sample (61.7%) were in support of such a plan.

### 3.4. Correlation between Demographics of the Participants and Their Knowledge and Attitudes about Air Pollution

Bivariate analysis was done to establish associations between the answers to the questionnaire items and the demographics of the sample population. Table 5 shows the distribution of the answers to the questionnaire among different subgroups of the sample. Only the statistically significant associations (*p*-value < 0.05) are reported.

### 3.5. Correlation between Knowledge of Physicians and Their Practices and Attitudes

Table 6 shows the associations between physicians’ knowledge (questionnaire items answered correctly) with their practices and attitudes towards air pollution. Only the statistically significant associations (*p*-value < 0.05) are reported.

## 4. Discussion

Air pollution has a tremendous impact on public health. Mitigating this effect requires controlling the sources of ambient air pollution, as well as reduction in indoor air pollution where people spend the majority of their time. Healthcare professionals have a big role in endorsing these solutions, first by being well informed on the sources of air pollution and then incorporating this information in their practice and daily interactions with their patients. The findings of this study show some knowledge deficit in the sample regarding air pollution.

Living near a busy road/intersection is a common source of ambient air pollution because people face a much higher exposure to traffic-related pollutants [9]. In our sample, only 60% of physicians agreed that living near an open and well-ventilated intersection is associated with air pollution. Moreover, Lebanon is located in a confined geographical area in the Middle East that is affected by dust storms from the Sahara Desert in Africa and the Arabian Desert in Asia [10]. When dust is transported away from its original source and is mixed with urban air, it elevates the levels of particulate matter in the air and becomes a source of air pollution. A study conducted in Beijing in 2004 showed that the mass concentration of PM_10_ was 5–10 higher during episodes of dust storms [11], and exposure to dust is associated with exacerbation of chronic respiratory diseases, such as COPD and asthma [12]. Only 52% of thephysicians in our survey categorized dust and dust storms as sources of air pollution. This underlines the need to educate Lebanese physicians on sources of ambient air pollution, including transported dust during dust storms, road intersections, and airports.

As far as indoor air pollution, most physicians agreed that cigarette smoke is a common air pollutant. However, the majority failed to recognize that vaping using electronic cigarette or smoking “heat-not-burn” devices such as IQOS are also sources of air pollution. In reality, the use of electronic smoking devices significantly increase the rates of indoor pollution [13]. Other controversial answers in our study were the smell of perfume, candles, vacuum cleaners, air fresheners, wood heaters, glues, and adhesives. Most responses did not consider these household items to be sources of air pollution. Meanwhile, a study conducted by Steinmann found that many of these consumer products (air fresheners, laundry products, cleaners, and personal care products) emit a range of volatile organic compounds (VOCs) that increase the rate of indoor air pollution [7]. In addition, staying indoors most of the time and living on lower level floors are also associated with increased exposure to indoor air pollution [14].

Another setting where physicians misjudged exposure to air pollution is their own workplace. Working in a hospital and in a classroom are both environments where air pollution can be abundant. A study conducted in Saudi Arabia to assess indoor air quality found that levels of particulate matter (both PM_10_ and TSP) in a university hospital were higher than the established air quality guidelines [15]. Moreover, indoor air pollution can be influenced by the crowdedness of each location, the characteristics of the building (e.g., ventilation rate), and the habits and activities of the visitors.

When it comes to the health impact of air pollution, there was an overwhelming agreement in our sample that air pollution increases the risk of cancer, COPD, asthma, and allergies. However, physicians were less knowledgeable about the effect that air pollution has on coronary artery disease, obesity, and insomnia. The Global Burden of Disease study estimated that pollution caused 9 million deaths in 2019; more than 60% were due to cardiovascular disease [16]. This is why it is important to implement training programs and counseling workshops for Lebanese physicians that address these associations with the aim of improving patient care.

Asking patients about exposure to air pollutants is not a common practice among physicians. Two thirds of our sample (61.7%) seldom asked their patients about their exposure to air pollution. If they did ask, the most frequent trigger was when patients reported specific symptoms related to air pollution (39.1%). Nevertheless, the majority of physicians (83.3%) did consider this question to be important, and 64.6% of them believe that they have a role in reducing air pollution both as physicians and citizens. Indeed, healthcare professionals have a substantial influence on the decision-making process when it comes to healthcare issues such as air pollution [9,17]. This can be done by publicly defending scientific evidence that shows the harmful effects of air pollution and calling for new policies to improve air quality in Lebanon and around the world.

The majority of physicians in our study (71.9%) expressed interest in counseling their patients on reducing their exposure to air pollution, but only half the sample (55.6%) were confident in their ability to do so. Additionally, two thirds of the sample (61.7%) expressed support of the application of a plan that includes assessment of patients’ exposure to air pollution as a regular part of medical assessment.

Our results are replicated in other studies, including a questionnaire conducted among to Polish physicians to assess their awareness of the danger of air pollution on health [18,19,20]. Only 25% of physicians believed that their knowledge on the healthcare impact of air pollution was sufficient. Approximately 5% knew what air pollution concentrations are acceptable, and 17% were up to date on air pollution levels in their region. Additionally, only 3% of the physicians in that study reported informing their patients when air pollution exceeded the guideline limits [18]. Rotily et al. also showed that only 44% of French physicians were concerned about air pollution. While all physicians knew the health effects of air pollution episodes on the respiratory tract, only half knew that mortality rates can increase significantly during such episodes. Approximately 40% of physicians had never heard about air pollution episodes that occurred in their city, while air quality control networks had identified them [5]. Another study conducted in Tehran showed that physicians with longer work experience had better performance on a questionnaire assessing their moral responsibility and the roles they can play in decreasing air pollution. Those who worked in offices or at universities also had better performances compared to those working in hospitals [21]. A qualitative, interview-based report was done in India to establish the status of knowledge, awareness, and practice of health care professionals with regard to air pollution and its impacts on health [22]. The main findings of the study were that air pollution is not currently an important topic of conversation within the community of physicians. In fact, air pollution was not part of their medical curriculum at all [22]. Therefore, it is important to incorporate changes in the medical training of physicians during medical school and to propose other strategies for currently practicing physicians, as well. Additionally, mitigating the effects of air pollution can be achieved by adapting strict air quality regulations and supporting them on an international level. Air pollution is a global issue, and sensitizing the physicians to its harmful impact should be carried out by international organizations. Another way to ensure that the burden of air pollution is appropriately addressed is to increase people’s awareness about measures they can take to protect themselves from air pollution.

Finally, the responses to our questionnaire were evaluated in terms of the demographic characteristics of the sample (Table 5). Practicing physicians in Beirut were more knowledgeable about several sources of air pollution (cars, incinerators, candles, and heaters) as compared to physicians outside Beirut. Moreover, female physicians were more likely to answer correctly on questions related to smoking and waste burning as compared to male physicians. As far as their attitudes, physicians above 40 years of age and those who had been practicing for more than 10 years were more confident in their ability to counsel patients and more in support of a national assessment plan for air pollution. Table 6 also shows the correlation between knowledge-practice and knowledge-attitude. The results show that physicians who were knowledgeable about sources of air pollution were more likely to ask their patients about air pollution exposure. They also believed that they had a role in reducing air pollution but were not confident in their ability to counsel patients about air pollution. Moreover, they were interested in attending continued medical education on air pollution and were in support of a national assessment plan for air pollution.

To our knowledge, this is the first study to associate the demographics of physicians with their knowledge and attitude towards air pollution and to show correlation between knowledge and practices and knowledge and attitudes. This is extremely important for policymakers because sensitizing physicians to air pollution should be targeted towards specific subgroups that lack the knowledge and would benefit the most from incorporating this issue into their continuous medical education.

Given that the response rate was only 14%, there is a high likelihood of selection bias in our study. Moreover, participants who are more knowledgeable and passionate about the issues discussed in the study were more likely to respond to the survey than those who are indifferent towards air pollution. Therefore, the sample is probably not representative of the general population of Lebanese physicians. Our survey questionnaire also lacked open-ended responses, which could have provided more in-depth analysis of the knowledge and attitudes of physicians towards air pollution. Future research could focus on developing stronger qualitative questions on the knowledge, attitudes, and practices of healthcare professionals regarding air pollution.

## 5. Conclusions

As Lebanon is experiencing higher levels of air pollution and given the constant accumulation of evidence on the health impact of exposure to air pollution, healthcare professionals need to stay up to date on this topic. The results of this study show that physicians lack knowledge about many sources of air pollution. Only 38% of them ask their patients about air pollution exposure, and only half of them are confident in air pollution counseling. This suggests the need for continuing education about air pollution for physicians and the development of guidelines for what exactly to ask patients in assessing their exposure. This necessity should also be extrapolated and dealt with on a global level because air pollution is a universal issue. International organizations should sensitize physicians to take an active role in preventing its harmful effects.

## Figures and Tables

**Figure 1 ijerph-19-07907-f001:**
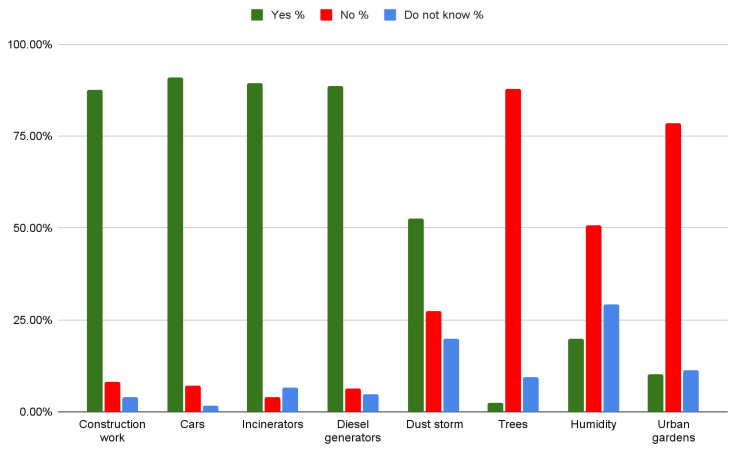
Answers to the question “Which of the below do you think are sources of air pollution”?

**Figure 2 ijerph-19-07907-f002:**
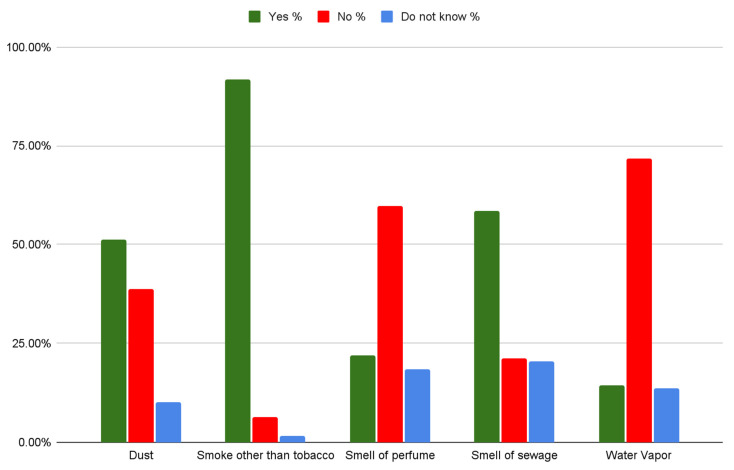
Answers to the question “Which of the below do you think are air pollutants”?

**Figure 3 ijerph-19-07907-f003:**
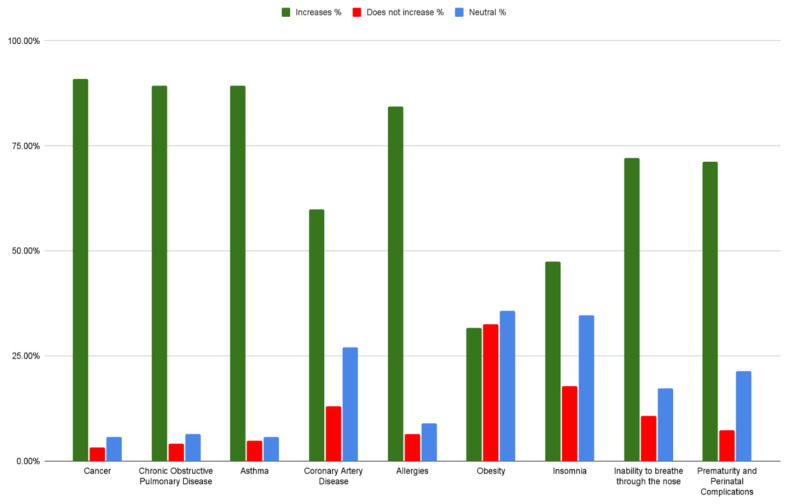
Answers to the question “To what extent do you think air pollution increases the risk of the following health conditions”?

**Table 1 ijerph-19-07907-t001:** Demographic characteristics of the sample (*n* = 125).

Characteristic	Frequency	Percent
Sex (*n* = 100)	52	52
Males		
Age (*n* = 112)		
<30 years	27	24.1
30–39 years	27	24.1
40–49 years	21	18.8
50 and older	37	33
Years of practice (*n* = 112)		
Less than 1 year	20	17.9
1–5 years	20	17.9
6–10 years	14	12.5
11–15 years	13	11.6
16–20 years	14	12.5
More than 20 years	31	27.7
Specialty (*n* = 112)		
Family medicine	27	24.1
General practice	27	24.1
Infectious disease	14	12.5
Cardiology	12	10.7
Pulmonary	11	9.8
Oncology	8	7.1
Pediatrics	6	5.4
Other (obstetrics, allergies, >1 specialty)	7	6.3
Non-medical degree	17	15.3
(Master’s in public health, epidemiology, medical teaching, clinical nutrition)		
Muhafaza (*n* = 109)		
Beirut	61	56
Mount Lebanon	25	22.9
North Lebanon	10	9.2
South Lebanon	9	8.3
Bekaa	4	3.7

**Table 2 ijerph-19-07907-t002:** Answers to the question “To what extent do you agree that each of the below living conditions is associated with exposure to air pollution”? The answers are reported as true counts (percentage).

Living Condition	Strongly Agree	Agree	Neutral	Disagree	Strongly Disagree
Living near an open and well-ventilated road intersection	24 (19.5)	50 (40.7)	25 (20.3)	16 (13.0)	8 (6.5)
Living near a gas station	48 (39.3)	48 (39.3)	20 (16.4)	6 (4.9)	0
Living near a diesel generator	91 (73.4)	20 (16.1)	7 (5.6)	3 (2.4)	3 (2.4)
Living near car or bus parking	64 (51.6)	47 (37.9)	9 (7.3)	3 (2.4)	1 (0.8)
Living near a construction site	55 (45.1)	43 (35.2)	14 (11.5)	6 (4.9)	4 (3.3)
Living in a house with a garden	14 (11.6)	11 (9.1)	6 (5.0)	33 (27.3)	57 (47.1)
Living near a garbage incinerator	72 (60.0)	26 (21.7)	11 (9.2)	4 (3.3)	7 (5.8)
Staying indoors most of the time	9 (7.3)	20 (16.3)	56 (45.5)	25 (20.3)	13 (10.6)
Living on higher-level floors (>third floor)	6 (5.0)	18 (14.9)	54 (44.6)	37 (30.6)	6 (5.0)
Living near a busy traffic intersection	65 (52.8)	38 (30.9)	9 (7.3)	8 (6.5)	3 (2.4)
Living near an open garbage dump	73 (59.3)	34 (27.6)	8 (6.5)	4 (3.3)	4 (3.3)
Smoking electronic cigarettes (e-cigarette, IQOS)	50 (41.7)	32 (26.7)	25 (20.8)	8 (6.7)	5 (4.2)
Smoking tobacco cigarettes	85 (68.5)	21 (16.9)	11 (8.9)	2 (1.6)	5 (4.0)
Smoking arguileh (hookah)	87 (70.7)	20 (16.3)	8 (6.5)	2 (1.6)	6 (4.9)
Living on the lower-level floors (<third floor)	10 (8.3)	44 (36.4)	56 (46.3)	10 (8.3)	1 (0.8)
Living near the airport	40 (32.5)	49 (39.8)	22 (17.9)	6 (4.9)	6 (4.9)
Living near an agricultural field	6 (5.0)	18 (14.9)	39 (32.2)	33 (27.3)	25 (20.7)
Living with a person who smokes	67 (54.0)	37 (29.8)	9 (7.3)	7 (5.6)	4 (3.2)

**Table 3 ijerph-19-07907-t003:** Answers to the question “To what extent do you agree that each of the below working conditions is associated with exposure to air pollution”? The answers are reported as true counts (percentage).

Work Condition	Strongly Agree	Agree	Neutral	Disagree	Strongly Disagree
Working at a gas station	67 (54.9)	38 (31.1)	12 (9.8)	2 (1.6)	3 (2.5)
Working in a place where smoking is NOT allowed	10 (8.1)	11 (8.9)	17 (13.8)	27 (22.0)	58 (47.2)
Working as a tour guide	11 (9.0)	19 (15.6)	51 (41.8)	28 (23.0)	13 (10.7)
Working at a clinic	8 (6.5)	15 (12.2)	38 (30.9)	45 (36.6)	17 (13.8)
Working in a hospital	8 (6.5)	20 (16.3)	37 (30.1)	47 (38.2)	11 (8.9)
Working in an automobile repair shop	40 (32.5)	58 (47.2)	16 (13.0)	8 (6.5)	1 (0.8)
Working at a gardening shop	4 (3.3)	18 (15.0)	30 (25.0)	41 (34.2)	27 (22.5)
Working at a construction site	51 (41.5)	50 (40.7)	15 (12.2)	7 (5.7)	0
Working in a chemical lab	42 (34.4)	50 (41.0)	18 (14.8)	7 (5.7)	5 (4.1)
Working with paints	64 (51.6)	42 (33.9)	9 (7.3)	5 (4.0)	4 (3.2)
Working in a place where smoking is allowed	70 (56.5)	35 (28.2)	9 (7.3)	8 (6.5)	2 (1.6)
Working as a teacher	6 (4.9)	21 (17.2)	38 (31.1)	34 (27.9)	23 (18.9)
Working in a place where only vaping (e.g., IQOS) is allowed	27 (23.3)	44 (37.9)	34 (29.3)	8 (6.9)	3 (2.6)
Working as a farmer	5 (4.2)	32 (26.9)	31 (26.1)	30 (25.2)	21 (17.6)

**Table 4 ijerph-19-07907-t004:** Answers to the question “To what extent do you agree that each of the below household items or activities is associated with exposure to air pollution”? The answers are reported as true counts (percentage).

Household Item	Strongly Agree	Agree	Neutral	Disagree	Strongly Disagree
Chimney	52 (42.6)	47 (38.5)	17 (13.9)	4 (3.3)	2 (1.6)
Candles	11 (9.1)	46 (38.0)	44 (36.4)	15 (12.4)	5 (4.1)
Wood heaters	32 (26.4)	56 (46.3)	23 (19.0)	10 (8.3)	0
Indoor smoking	91 (74.0)	22 (17.9)	5 (4.1)	2 (1.6)	3 (2.4)
Detergents [Odex or Flash]	56 (45.5)	43 (35.0)	14 (11.4)	10 (8.1)	0
Watching television	7 (5.9)	15 (12.6)	29 (24.4)	29 (24.4)	39 (32.8)
Air conditioning unit	6 (5.0)	34 (28.3)	45 (37.5)	28 (23.3)	7 (5.8)
Vacuum cleaners	9 (7.4)	31 (25.6)	47 (38.8)	23 (19.0)	11 (9.1)
Pets	7 (5.7)	31 (25.4)	37 (30.3)	29 (23.8)	18 (14.8)
Indoor cooking	4 (3.3)	31 (25.4)	42 (34.4)	31 (25.4)	14 (11.5)
Pesticides and insecticides	69 (56.1)	33 (26.8)	12 (9.8)	7 (5.7)	2 (1.6)
Mold and fungi on the walls	53 (43.1)	37 (30.1)	23 (18.7)	9 (7.3)	1 (0.8)
Air fresheners	10 (8.4)	42 (35.3)	46 (38.7)	17 (14.3)	4 (3.4)
Glues/adhesives	27 (22.1)	43 (35.2)	34 (27.9)	12 (9.8)	6 (4.9)
Taking a shower	3 (2.5)	14 (11.6)	21 (17.4)	35 (28.9)	48 (39.7)
Open windows at home	9 (7.4)	15 (12.3)	35 (28.7)	29 (23.8)	34 (27.9)
Lighting charcoal for arguileh at home	71 (57.7)	33 (26.8)	12 (9.8)	4 (3.3)	3 (2.4)
Diesel heaters indoors	73 (59.3)	28 (22.8)	10 (8.1)	8 (6.5)	4 (3.3)

**Table 5 ijerph-19-07907-t005:** Distribution of the answers among the different subgroups of the sample, divided according to demographics.

Questionnaire Items	Demographic Characteristics		*p*-Value
	Muhafaza		
	Beirut	Outside Beirut	
Cars	58 (95.1%)	39 (83.0%)	0.039
Incinerators	57 (95.0%)	37 (78.7%)	0.011
Household candles	33 (54.1%)	15 (31.9%)	0.021
Household wood heaters	51 (83.6%)	26 (56.5%)	0.002
Indoor diesel heaters	54 (88.5%)	34 (72.3%)	0.032
Risk of developing cancer	58 (96.7%)	39 (83.0%)	0.016
	Sex		
	Male	Female	
Living near a garbage incinerator	36 (70.6%)	40 (90.9%)	0.014
Smoking tobacco	39 (75.0%)	44 (93.6%)	0.012
Living with a person who smokes	40 (76.9%)	43 (91.5%)	0.049
Household detergents	45 (86.7%)	33 (70.2%)	0.047
	Age		
	Under 40 years	40 years and older	
Living near a gas station	37 (68.5%)	49 (86.0%)	0.028
Confidence in counseling patients	23 (42.6%)	38 (65.5%)	0.015
Support of national assessment plan	20 (37.0%)	48 (82.8%)	0.000
	Years of Practice		
	Under 10 years	10 years and more	
Confidence in counseling patients	23 (42.6%)	38 (65.5%)	0.015
Support of national assessment plan	21 (38.9%)	47 (81.0%)	0.000
	Specialty		
	Internal medicine	Other specialties	
Dust storms	29 (64.4%)	27 (42.9%)	0.027
Support of national assessment plan	37 (80.4%)	31 (47.0%)	0.000

**Table 6 ijerph-19-07907-t006:** Distribution of answers on air pollution knowledge according to the practices and attitudes of physicians.

Questionnaire Items on Knowledge	Questionnaire Items on Practices and Attitudes		*p*-Value
	Asking patients about their exposure to air pollution		
	Yes	No	
Construction work	82 (91.1%)	19 (76.0%)	0.041
Diesel generators	84 (92.3%)	18 (72.0%)	0.006
Smoking, other than tobacco	85 (94.4%)	20 (80.0%)	0.023
Smoking arguileh (hookah)	82 (91.1%)	18 (72.0%)	0.012
Working with paint	82 (90.1%)	17 (68.0%)	0.006
Working where vaping is allowed	61 (70.9%)	6 (24.0%)	0.000
Pesticides and insecticides	79 (86.8%)	17 (68.0%)	0.027
Increases risk of CAD	60 (65.9%)	10 (40.0%)	0.019
Increases risk of severe illness	73 (80.2%)	14 (56.0%)	0.013
	Importance of asking patients about air pollution exposure		
	Important	Not important	
Incinerators	85 (91.4%)	14 (73.7%)	0.028
Smoking arguileh (hookah)	84 (90.3%)	14 (73.7%)	0.046
	Role in reducing air pollution		
	Yes	No	
Construction work	96 (89.7%)	3 (60.0%)	0.043
Living on lower-level floors	47 (44.3%)	0 (0.0%)	0.050
Living near an airport	81 (75.7%)	1 (20.0%)	0.006
	Confidence in counseling patients on air pollution		
	Yes	No	
Living near a traffic intersection	49 (77.8%)	46 (92.0%)	0.040
Increases risk of cancer	53 (85.5%)	50 (98.0%)	0.019
Increases risk of asthma	53 (84.1%)	49 (96.1%)	0.039
	Interest in attending continuing education about air pollution		
	Yes	No	
Living near a gas station	68 (84.0%)	20 (62.5%)	0.013
Living on lower-level floors	39 (48.8%)	9 (28.1%)	0.046
Living near an airport	65 (80.2%)	17 (53.1%)	0.004
Working at a gas station	73 (91.3%)	23 (71.9%)	0.008
	Support of national air pollution assessment plan		
	Yes	No	
Dust storms	41 (60.3%)	16 (37.2%)	0.018
Living near diesel generator	66 (94.3%)	36 (81.8%)	0.035
Living near garbage incinerator	60 (89.6%)	30 (69.8%)	0.009
Smoking tobacco	65 (91.4%)	33 (75.0%)	0.017
Living near an airport	57 (81.4%)	26 (59.1%)	0.009
Working where smoking is allowed	63 (90.0%)	33 (75.0%)	0.033
Household mold and fungi on walls	55 (78.6%)	27 (61.4%)	0.047
Increases the risk of allergies	63 (90.0%)	33 (75.0%)	0.033
Increases the risk of severe illness	58 (82.9%)	27 (61.4%)	0.010

## Data Availability

The data supporting the conclusion of this article are available upon request from the corresponding author.

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
