# Peer review of "A Survey on the Knowledge, Attitudes, and Practices of Lebanese Physicians Regarding Air Pollution"

_ijerph, 2022, doi:10.3390/ijerph19137907_

Round 1

Reviewer 1 Report

This manuscript titled "A survey on the knowledge, attitudes, and practices of Lebanese physicians towards air pollution" was reviewed in detail, questionnaire was adpot to collect the information about knowledge, attitudes and practices from doctors to understand the literacy of air quality with human health. The topic is interesting, wording is good, I understand the research is important after reading, but for official publication, there are still two major concepts need to improve and clarify, I suggest authors have to answer my first concern and supply my second suggestion in the following: 

  1. About the main purpose of the manuscript, knowledge, atttitudes and practices are three sectors to examine literacy which are already well-design in education fields. They will interact with each other, from the manuscript we only learn from each independent part, that will offer misunderstanding in this field. For example, how about the knowledge interact with practice? a well-knowledged doctor can have good attitude about air pollution? and he/she can treat the pacient with the knowledge? I can not have any idea from the discussion section. For the reason, I do suggest authors should do more statistic analysis about the interaction, the situation for now, the analysis is too simple to be published.
  2. As the survey sampling, I would like to have more information aroud the study area, not only the information about the samples but also the environment and background. These information are quite important to check the confidence of this manuscript, just how many physisians totally in Lebanon who take care about air pollution related events, the environment/indoor air quality information in detail to supply the emission amount, air pollutant concentration statistic (long term average) results...etc. These information can link the sitation for one country to international communuties, will raise the application of the manuscript to global society.  

Author Response

Reviewer 1:

This manuscript titled "A survey on the knowledge, attitudes, and practices of Lebanese physicians towards air pollution" was reviewed in detail, questionnaire was adpot to collect the information about knowledge, attitudes and practices from doctors to understand the literacy of air quality with human health. The topic is interesting, wording is good, I understand the research is important after reading, but for official publication, there are still two major concepts need to improve and clarify, I suggest authors have to answer my first concern and supply my second suggestion in the following: 

1. About the main purpose of the manuscript, knowledge, attitudes and practices are three sectors to examine literacy which are already well-design in education fields. They will interact with each other, from the manuscript we only learn from each independent part, that will offer misunderstanding in this field. For example, how about the knowledge interact with practice? a well-knowledged doctor can have good attitude about air pollution? and he/she can treat the pacient with the knowledge? I can not have any idea from the discussion section. For the reason, I do suggest authors should do more statistic analysis about the interaction, the situation for now, the analysis is too simple to be published.

We agree with you. A new section is added to correlate physicians’ knowledge with their practices and attitudes towards air pollution.  Please refer to page 11 section 4 and 5 “Correlation between demographics of the participants and their knowledge and attitudes towards air pollution” and “Correlation between knowledge of physicians with their practices and attitudes” respectively.

2.  As the survey sampling, I would like to have more information aroud the study area, not only the information about the samples but also the environment and background. This information are quite important to check the confidence of this manuscript, just how many physisians totally in Lebanon who take care about air pollution related events, the environment/indoor air quality information in detail to supply the emission amount, air pollutant concentration statistic (long term average) results...etc. This information can link the sitation for one country to international communities, will raise the application of the manuscript to global society.

Thank you for your comment. Unfortunately, there is a severe lack of studies measuring levels of air pollution in Lebanon and surrounding region. Only one study measured the levels of PM2.5 and PM10 in Beirut and it was discussed in the introduction of this paper (page 3 line 89).

Reviewer 2 Report

Attached please find the comments.

Author Response

Reviewer 2:

The manuscript titled “A survey on the knowledge, attitudes, and practices of Lebanese physicians towards air pollution” is well written and provides data support for Labanese physicians' attitudes towards air pollution.

The mechanism reported in this manuscript is significant. I recommend this manuscript be published with minor revision:

(1) There are too few sample quantities in this article to cause data to deviation. It is recommended to expand the sample selection range in subsequent work to get more accurate data support.

Thank you for your comment. We agree with you and will work on it in future studies.

(2) In the conclusion, some quantitative data should be written to show the correctness of the conclusion more intuitively.

Quantitative data are now added to the conclusion. Please refer to page 15 line 390.

(3) The improvements mentioned in the article should not only consider the knowledge and attitude of doctors on air pollution, but also increased people's awareness of their own protection for air pollution.

Thank you for your comment. We have mentioned this in the discussion part. Please refer to page 15.

(4) There is a format error in the reference in the reference. Please modify it.

Reference part was modified.

(5) Some new literatures might be helping the authors to further deepen the understanding of reaction mechanism as well as newest developing in this field (Bioresource Technology, 2021, 332: 125086 Study on the Hg0 removal characteristics and synergistic mechanism of iron-based modified biochar doped with multiple metals).

We believe that this paper is of tremendous relevance, however, we could not identify in what aspect could this study be incorporated in our article.

Reviewer 3 Report

General comment

A very interesting and original study, in a sensitive geographical region. Ideas and especially misconceptions of physicians on air quality are very important for primary care, yet they are rarely studied. Huge benefits could be anticipated by having physicians trained in air pollution effects on public health. The paper is direct in its message and easy to read. Some technical improvements are needed (e.g. evaluation of the responses by demographic characteristics of the sample, better statistical analysis, comparisons with literature). I list below some comments in this direction.

Specific comments

Lines 42-59: Cite some of the major epidemiological studies that solidified the effects of air pollution on mortality and cardiorespiratory outcomes (short- and long-term) and add some discussion about the effects for which exposure to air pollution was verified to have causality (not just probable or suspected causality).

Line 62: Granular is not appropriate here, especially for liquid aerosols. Rephrase.

Line 66: Maybe you want to indicate the WHO estimates for mortality due to indoor exposure that is almost on par with the ambient exposure (see WHO website for current estimates).

Lines 67: “Fumes of burned garbage” could be rephrased as “waste burning”.

Line 69: PM2.5 and PM10 must have the 2.5 and 10 as subscripts. Check throughout the text.

Line 70: Are there any local air quality standards enforced in Lebanon?

Line 77: Already established, remove sentence and rephrase the next.

Section 2: There is no mention whether statistical indicators were used to assess the difference in observed contrasts and variations (in the figures I can see only percentages, this is relatively basic as an analysis). Also it not clear how the responses were evaluated in terms of the demographic characteristics of the sample presented in Fig. S1. Factors like age or specialty might be very important as they could imply prior misconceptions. Please augment your analysis taking into account these points.

 Line 93: Since the questionnaire is not provided in the appendix, it has to be described in more detail here.

Section 3: I can’t see any Figures or Table in the main text. This is somewhat peculiar for a research paper. I suggest including Table S1 and Figures S1, S2, S3 in the manuscript.

Lines 208-211: Move to introduction

Lines 218-219: Reword this, not clear what you mean.

Line 235: What are shishas? Seems to be a specialized term, please explain or write something like “electronic smoking devices”.

Line 273: Start a new paragraph

Lines 273-285: Are there any similar studies in Western Europe or the US? Comparisons would permit to identify ethnographical patterns or differences in academic training of physicians. For example, in most European medical schools, future physicians become introduced to air quality issues in their public health and epidemiology classes (probably also in pulmonology and cardiology). However, the effect of such training on their future attitudes against air pollution and how they include it in their practice is unclear. It would be interesting if you would discuss this aspect.

 Line 295-299: Structured interviews are also an interesting and much studied - from a scientific standpoint – facet of quantitative research. Maybe you could consider also this in future research, given also the relatively low response rate in the questionnaire survey (i.e. to enhance the obtained information).

Author Response

Reviewer 3:

General comment

A very interesting and original study, in a sensitive geographical region. Ideas and especially misconceptions of physicians on air quality are very important for primary care, yet they are rarely studied. Huge benefits could be anticipated by having physicians trained in air pollution effects on public health. The paper is direct in its message and easy to read. Some technical improvements are needed (e.g. evaluation of the responses by demographic characteristics of the sample, better statistical analysis, comparisons with literature). I list below some comments in this direction.

Specific comments

Lines 42-59: Cite some of the major epidemiological studies that solidified the effects of air pollution on mortality and cardiorespiratory outcomes (short- and long-term) and add some discussion about the effects for which exposure to air pollution was verified to have causality (not just probable or suspected causality).

Thank you for your comment. We have added some new references that correlate air pollution with cardiovascular disease.

Line 62: Granular is not appropriate here, especially for liquid aerosols. Rephrase.

We agree with you. We have changed “Granular” into “present”.

Line 66: Maybe you want to indicate the WHO estimates for mortality due to indoor exposure that is almost on par with the ambient exposure (see WHO website for current estimates).

Thank you for your comment. We have added in the introduction the mortality estimates due to indoor (household) pollution. Please refer to page 3 line 59.

Lines 67: “Fumes of burned garbage” could be rephrased as “waste burning”.

Agree. We have rephrased it accordingly. Please refer to line 81 page 3.

Line 69: PM2.5 and PM10 must have the 2.5 and 10 as subscripts. Check throughout the text.

Thank you for your comment. We have modified it as subscripts accordingly in the manuscript.

Line 70: Are there any local air quality standards enforced in Lebanon?

Unfortunately, to date, there is no air quality standards in Lebanon.

Line 77: Already established, remove sentence and rephrase the next.

Thank you for your comment. We have removed and rephrased the sentence accordingly. Please refer to page 3 line 90. 

Section 2: There is no mention whether statistical indicators were used to assess the difference in observed contrasts and variations (in the figures I can see only percentages, this is relatively basic as an analysis). Also it not clear how the responses were evaluated in terms of the demographic characteristics of the sample presented in Fig. S1. Factors like age or specialty might be very important as they could imply prior misconceptions. Please augment your analysis taking into account these points.

We agree with you. Hence, additional statistics using Chi-square were added to correlate demographics of the sample with their knowledge, attitude, and practices towards air pollution. Please refer in the results section, section 4, page 11 and table 5.

 Line 93: Since the questionnaire is not provided in the appendix, it has to be described in more detail here.

We believe that the questionnaire should be added to the appendix. We did accordingly. Refer to page 17 “Appendix” section.

Section 3: I can’t see any Figures or Table in the main text. This is somewhat peculiar for a research paper. I suggest including Table S1 and Figures S1, S2, S3 in the manuscript.

Tables and figures are now added to the manuscript. We have previously put them in the “supplementary material, however we believe that they should be integrated in the manuscript.

Lines 208-211: Move to introduction

Done.

Lines 218-219: Reword this, not clear what you mean.

Done.

Line 235: What are shishas? Seems to be a specialized term, please explain or write something like “electronic smoking devices”.

We have changed the term “shishas” to “electronic smoking devices”.

Line 273: Start a new paragraph

Done.

Lines 273-285: Are there any similar studies in Western Europe or the US? Comparisons would permit to identify ethnographical patterns or differences in academic training of physicians. For example, in most European medical schools, future physicians become introduced to air quality issues in their public health and epidemiology classes (probably also in pulmonology and cardiology). However, the effect of such training on their future attitudes against air pollution and how they include it in their practice is unclear. It would be interesting if you would discuss this aspect.

Thank you for your comment. We have added in the discussion section similar studies conducted in Poland, France, India, and Tehran. Please refer to page 14 lines 335 and 344.

 Line 295-299: Structured interviews are also an interesting and much studied - from a scientific standpoint – facet of quantitative research. Maybe you could consider also this in future research, given also the relatively low response rate in the questionnaire survey (i.e. to enhance the obtained information).

We agree with you. We have mentioned this in the limitations section and will be noted for future research.

Reviewer 4 Report

This paper conducts a survey on the knowledge, attitudes, and practices of physicians towards air pollution. Overall, the topic of this paper is helpful. However, the biggest flaw of the study is the lack of depth of the data analysis. The reviewer has the following comments that may help improve the manuscript.

  • While there is a lack of studies that addresses the topic in Lebanon, there is a lack of discussion on existing studies of other places. For example, what are the major findings or implications of these studies?
  • There is a lack of justification of the study design. That is, how the authors developed the design?
  • The time period of the study is not mentioned at all.
  • The IRB information (e.g., IRB approval proof) is missing.
  • The instrument of the study questionnaire needs to be justified.
  • Why only five physicians were invited to conduct the pilot test? What were the major revision based on such test?
  • Why were all the figures/tables included in the SI instead of the main manuscript?
  • It seems that the entire manuscript relied only on the descriptive analysis. Have the authors conducted other tests?

Author Response

Reviewer 4:

This paper conducts a survey on the knowledge, attitudes, and practices of physicians towards air pollution. Overall, the topic of this paper is helpful. However, the biggest flaw of the study is the lack of depth of the data analysis. The reviewer has the following comments that may help improve the manuscript.

  • While there is a lack of studies that addresses the topic in Lebanon, there is a lack of discussion on existing studies of other places. For example, what are the major findings or implications of these studies?

We agree with you. We have added in the discussion section several comparative studies conducted in Poland, France, India and Tehran, similar to our study. Kindly refer to page 14 lines 335 and 344.

  • There is a lack of justification of the study design. That is, how the authors developed the design?

Thank you for your comment. Given that the aim of the study was to describe the knowledge, attitude and practices of physicians related to air pollution, the authors chose the descriptive design since there is no intervention or manipulation of the variables of interest. The variables were studied as they occur. The cross-sectional correlational approach is justified by the aim of identifying associations between characteristics of the sample (age, gender and years of experience) and their knowledge, attitude and practices; the investigators were examining associations not causation, thus justifying the selected design. A self-administered survey approach was used since knowledge, attitudes and practices are best measured from the perspective of the studied participants, in this case the physicians. In order to reach out to physicians we designed the study using the specialty medical societies from where we got their email addresses, in order to send them the link to the online survey, which was the method of data collection.

  • The time period of the study is not mentioned at all.

The study was conducted between October 2020 and January 2021. We have mentioned this in the methodology section.

  • The IRB information (e.g., IRB approval proof) is missing.

The IRB protocol code and date of approval are mentioned in the IRB statement at the end of the manuscript.

  • The instrument of the study questionnaire needs to be justified.

The authors did not find a standard instrument in the literature that measures knowledge, attitude and practices related to air pollution. Each of the studies reviewed developed its own questionnaire. For that, the investigators developed their own questionnaire based on what is available in the literature, and tailored to answer the specific aims of the study and variables of interest, as well as taking into consideration the air pollution situation in the country where the study was conducted. Thus the sections of the questionnaire addressed knowledge of sources of air pollutions and what substances are considered to be pollutants, the attitude of the physicians regarding the importance of air pollution and getting training about this topic, as well as inclusion of assessment of exposure to air pollution in their patients and counseling their patients about air pollution in their practice.

  • Why only five physicians were invited to conduct the pilot test? What were the major revision based on such test?

As mentioned in the manuscript, the panel of experts who developed the questionnaire included 2 experts in air pollution (an engineer and a chemist), a cardiologist, a pulmonary and allergy specialist, and an oncologist, in addition to a professor of nursing with experience in developing surveys. The purpose of the pilot test was to make sure the questions were clear and relevant to the physicians. The pilot study was not planned to do reliability or validity testing of the questionnaire, so 5 physicians were considered enough to meet its aim.

  • Why were all the figures/tables included in the SI instead of the main manuscript?

We agree with you. The figures and tables are now in the main manuscript.

  • It seems that the entire manuscript relied only on the descriptive analysis. Have the authors conducted other tests?

Thank you for your comment. Additional statistics using Chi-square were added to correlate demographics of the sample with their knowledge on air pollution. Please refer to results section, tables 5 and 6 respectively.

Round 2

Reviewer 4 Report

The reviewer thanks the authors for resolving all comments. It would also be great to include more studies in the introduction and discussion. Overall, great work!